# Direct mechanical stimulation of tip links in hair cells through DNA tethers

**Aakash Basu, Samuel Lagier[†], Maria Vologodskaia, Brian A Fabella, AJ Hudspeth***

Laboratory of Sensory Neuroscience, Howard Hughes Medical Institute, The Rockefeller University, New York, United States

**Abstract** Mechanoelectrical transduction by hair cells commences with hair-bundle deflection, which is postulated to tense filamentous tip links connected to transduction channels. Because direct mechanical stimulation of tip links has not been experimentally possible, this hypothesis has not been tested. We have engineered DNA tethers that link superparamagnetic beads to tip links and exert mechanical forces on the links when exposed to a magnetic-field gradient. By pulling directly on tip links of the bullfrog's sacculus we have evoked transduction currents from hair cells, confirming the hypothesis that tension in the tip links opens transduction channels. This demonstration of direct mechanical access to tip links additionally lays a foundation for experiments probing the mechanics of individual channels.

***For correspondence:** hudspaj@rockefeller.edu

**Present address:** [†]Department of Basic Neurosciences, School of Medicine, University of Geneva, Geneva, Switzerland

**Competing interests:** The authors declare that no competing interests exist.

## Introduction

Hair cells occur universally in the auditory and vestibular systems of vertebrates, where they transduce mechanical stimuli into electrical responses and thus initiate the perception of sounds and accelerations (*Hudspeth, 2008*, *2014*). The apical surface of a hair cell bears a hair bundle comprising actin-filled stereocilia whose gradation in height forms a bevel (*Figure 1A*). The tip of each stereocilium is connected to the side of the tallest adjacent stereocilium through a protein filament, the tip link (*Pickles et al., 1984*). The upper two-thirds of this link comprises a parallel dimer of cadherin 23 (CDH23) molecules whereas the lower third encompasses a parallel dimer of protocadherin 15 (PCDH15) molecules (*Kazmierczak et al., 2007*).

Deflection of a hair bundle toward its tall edge opens mechanoelectrical-transduction channels that allow an influx of cations, predominantly $K^+$ but also $Ca^{2+}$, and thus depolarizes the hair cell. Electrophysiological and mechanical evidence suggests that deflection increases the tension in an elastic structure, the gating spring, that communicates force to the transduction channels (*Corey and Hudspeth, 1983a*). Several lines of circumstantial evidence support the hypothesis that the tip link constitutes at least a portion of the gating spring: the stereociliary tips are the site of transduction (*Hudspeth, 1982*; *Beurg et al., 2009*); the orientation of the tip links corresponds to the hair bundle's axis of maximal sensitivity (*Shotwell et al., 1981*); and the responsiveness vanishes when the tip links are disrupted (*Assad et al., 1991*). Nevertheless, the critical role of tip-link tension in gating mechanoelectrical-transduction channels has never been tested directly.

Because they occur in the narrow spaces between the tips of contiguous stereocilia, tip links are not readily accessible to experimental manipulation. As a consequence, the stimulation of transduction channels has heretofore been limited to the deflection of entire hair bundles. We have developed a method of applying mechanical force specifically to tip links, allowing us to test directly the hypothesis that tip-link tension gates the transduction channels.

**eLife digest** In animals with backbones, the inner ear is responsible for both hearing and balance. Sound waves and head movements apply a mechanical force to hair cells inside the inner ear. This causes the cells to produce electrical signals that ultimately communicate information about the sound or movement to the brain. The apparatus that converts mechanical forces into electrical signals is called the hair bundle, which is an upright cluster of small rods called stereocilia that protrude from the hair cell's flattened top surface. Fine filaments called tip links connect the stereocilia within a hair bundle to one another.

It is thought that the mechanical deflection of a hair bundle tenses the tip links and opens ion channels – molecular pores through which ions can pass – that are attached to the tip links. The resultant flow of ions across the hair cell's membrane would then cause a voltage change that in turn triggers the cell's electrical response. It has not been possible to test this hypothesis, however, because the position of the tip links within a hair bundle prevents them from being stimulated directly in experiments.

Basu et al. have now used specific antibody molecules to attach tip links to magnetic beads using a strand of DNA. The DNA acted as a string that penetrated into the hair bundles, connecting the tip links to magnetic beads outside the bundles. This meant that moving the bead by applying a magnetic force to it pulled upon the tip links, and the investigators observed that this activated the associated ion channels. The resultant electrical signals confirmed that tip links play a role in the responses of hair cells.

Although there are methods that allow the electrical activity from a single ion channel to be recorded, the new approach provides an opportunity for studying the mechanical activity of a channel as well. Future studies could therefore investigate the mechanical and electrical signals associated with individual tip links and the ion channels to which they attach in order to investigate the specific role they play in hearing.

## Results

To apply tension at a defined position on each of a large number of tip links, we raised a polyclonal antiserum against a specific epitope near the carboxyl terminal of the bullfrog's CDH23, and thus at the links' upper ends. Because each tip link resides in a confined space at the hair bundle's top, it is impractical to ligate the antibodies directly to a probe particle for use with optical or magnetic tweezers. We therefore created a mechanical tether for the tip link by interposing a 3 kb, double-stranded DNA molecule between the attached antibody and a 1 μm-diameter superparamagnetic bead (*Figure 1B*). The DNA molecule was obtained by PCR of a plasmid with primers that incorporated biotin and fluorescein moieties at the molecule's opposite ends. The biotinylated terminus bound through a streptavidin molecule to a biotinylated anti-rabbit IgG, which in turn bound the rabbit anti-CDH23. The fluorescein-modified end of the DNA adhered to a superparamagnetic bead coated with anti-fluorescein. The tether provided sufficient separation between the tip link and the relatively large bead to overcome the problem of steric inaccessibility.

Control experiments confirmed that the DNA tethers functioned as intended. First, the anti-CDH23 antibodies labeled the stereociliary tips in living hair cells of the bullfrog's sacculus (*Figure 1C,D*). The antibodies' binding to cadherin isoforms was also confirmed for both saccular and brain tissue by Western blot analysis. Following synthesis of the complete tether, superparamagnetic beads localized to the regions of the tip links but were absent if the primary antibody, secondary antibody, or streptavidin was omitted during the process of tether synthesis (*Figure 1E,F*). Finally, as expected for DNA tethers, treatment of the preparation with DNAse liberated the superparamagnetic beads.

The otolithic membrane that covered the hair bundles was removed enzymatically, exposing the tip links to the components of the DNA tethers. After completing tether synthesis we placed the chamber beneath an electromagnet and used magnetic force to pull the tethered beads directly upward, parallel to the stereocilia. If an increase in tip-link tension opens mechanoelectrical-transduction channels, pulling the links upward should elicit ionic currents into the hair cells and thus

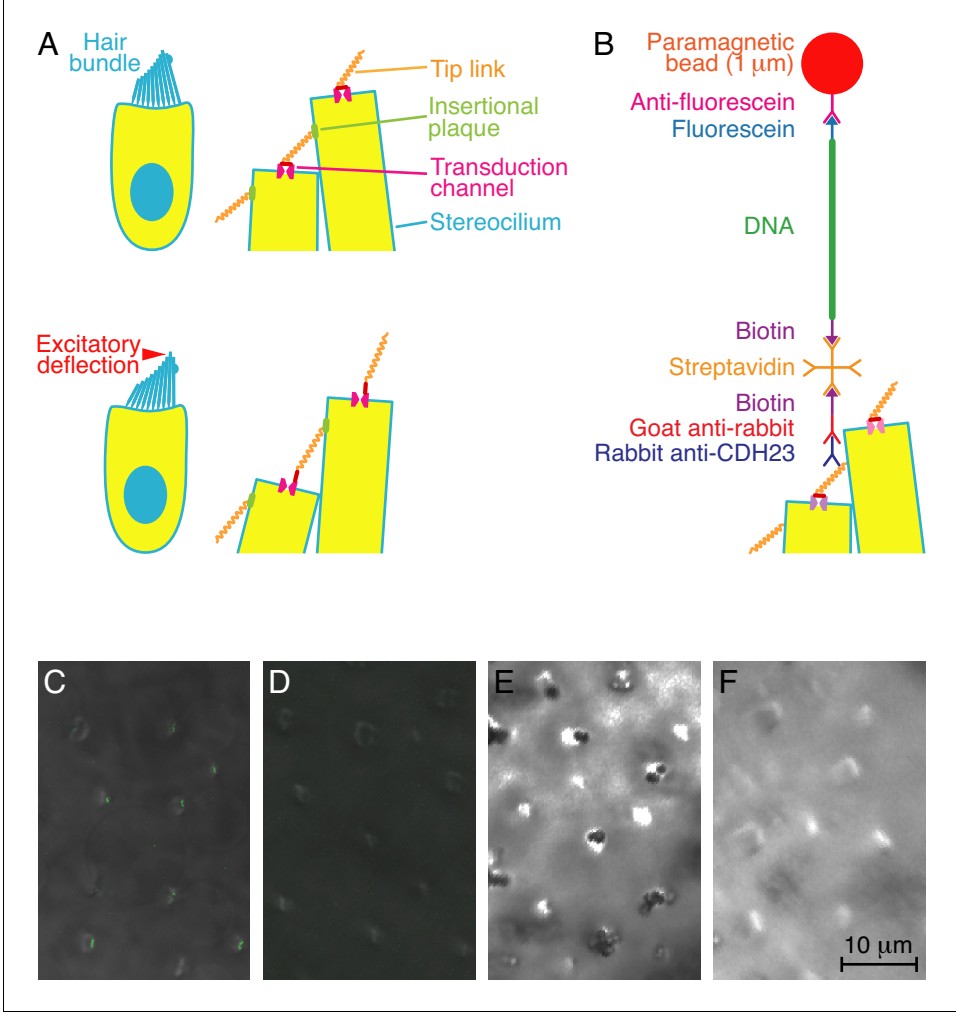

**Figure 1.** Experimental configuration and control experiments. (**A**) In diagrams of a hair cell at rest (top) and during excitatory stimulation (bottom), the magnified images show how hair-bundle deflection is thought to raise the tension in tip links and thus open transduction channels. (**B**) A diagram, not to scale, portrays the molecular assembly that tethers a superparamagnetic bead to a tip link. (**C**) In overlaid fluorescence and differential-interference-contrast images of the apical surface of a bullfrog's sacculus, anti-CDH23 immunolabels the tops of hair bundles. (**D**) Labeling is absent under otherwise identical conditions when the primary antibody is omitted. (**E**) A differential-interference-contrast micrograph of a sacculus shows clusters of DNA-tethered superparamagnetic beads atop many hair bundles. (**F**) No beads are present in a control preparation lacking the primary antibody.

produce a transepithelial electrical response termed the microphonic potential (*Corey and Hudspeth, 1983b*). Detected with electrodes in the two compartments of a recording chamber, such a signal reflects the total transduction current traversing the saccular epithelium (*Figure 2A*).

Upon the application of current through the electromagnet in either direction, we observed electrical responses whose polarity was consistent with inward cationic currents through the apical surfaces of the hair cells (*Figure 2B*). To confirm that these signals originated primarily from transduction channels, we conducted control experiments in the presence of 1 mM gentamicin, a blocker of transduction channels (*Kroese et al., 1989*). We then detected only small, magnetically induced artifacts during stimulation of either polarity. Moreover, no response was observed when the primary antiserum was omitted from the preparative procedure.

As a result of the electromotive force induced by the changing magnetic field, we expected our recordings to include stimulus artifacts. Because the pulling force on a superparamagnetic bead does not depend on the direction of the current through the electromagnet, the physiological

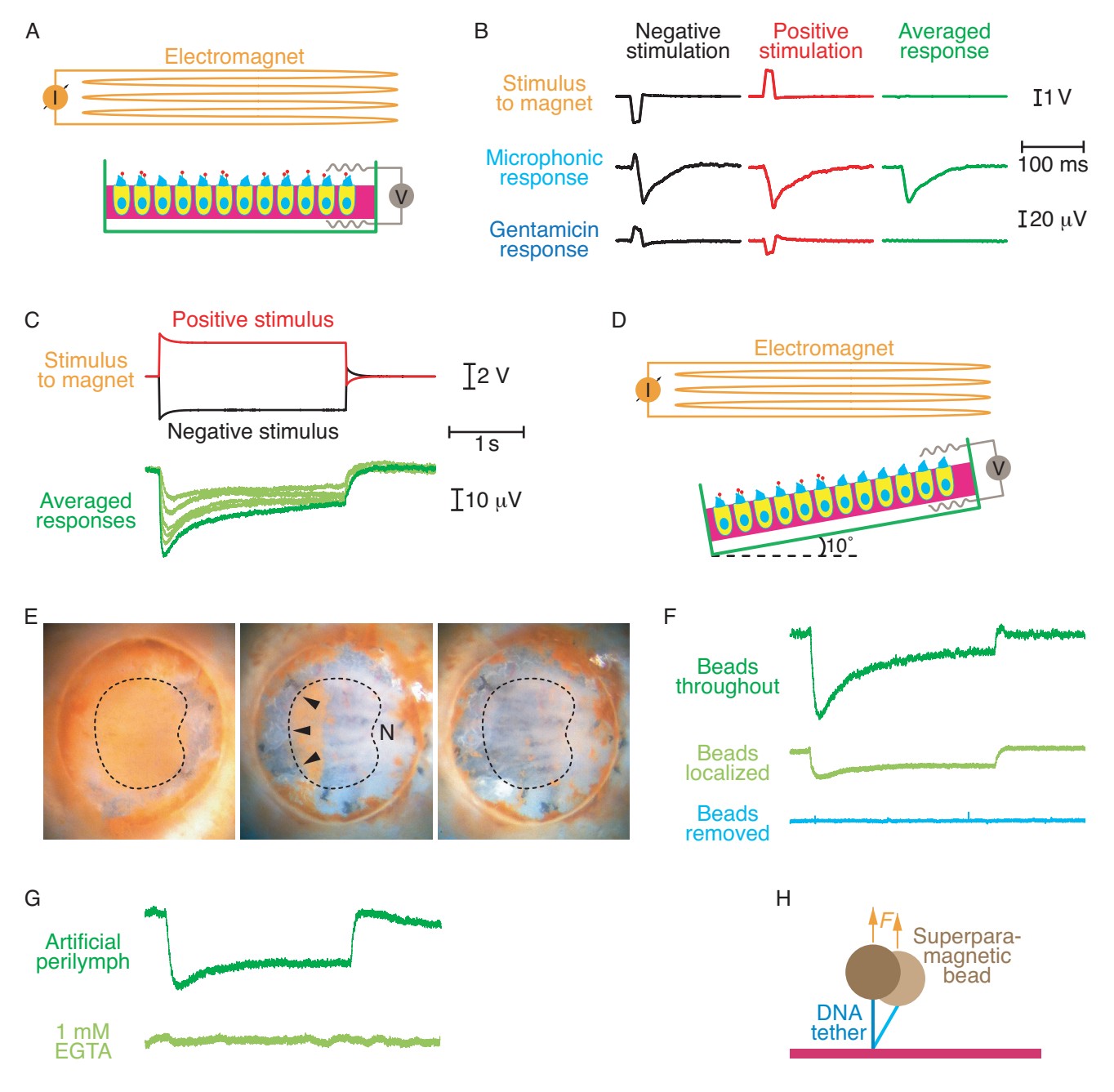

**Figure 2.** Electrical responses to magnetic stimulation. (**A**) The diagram shows a sacculus mounted in a two-compartment chamber beneath an electromagnet. The otolithic membrane has been removed and superparamagnetic beads (red dots) are attached to the tip links through DNA tethers. (**B**) Brief magnetic stimuli of opposite polarities, which are denoted by the voltages applied to the magnet, elicit transepithelial electrical responses. Exposure of the preparation to 1 mM gentamicin blocks these physiological responses, leaving only artifacts that are eliminated by averaging the signals obtained from stimuli of opposite polarity. (**C**) Longer stimuli demonstrate the persistence of transduction currents without the extensive adaptation observed during conventional stimulation. The stimulus traces shown produced the largest response; successively smaller responses were generated by lowering the voltage supplied to the magnet in steps of 0.2 V. (**D**) To eliminate the possibility that magnetic stimulation pulls hair bundles in the excitatory direction, most of the beads can be removed and the preparation tilted by 10°. (**E**) Brown superparamagnetic beads initially cover the apical surface of a mounted sacculus (left). Scraping away most of the beads (center) leaves only the beads attached to hair bundles of similar orientations (arrowheads), sensitive to stimulation away from the saccular nerve (N). Further manipulation removes nearly all the beads (right). The dashed line circumscribes the region in which hair cells occur. (**F**) Although smaller than the control response (top), an electrical response nevertheless persists after removal of most beads (middle). With the beads localized to one region with similar hair-bundle orientations, magnetic force pulls those bundles upward and toward their short edges. Removing the remaining beads eliminates the response (bottom). The calibration bars in (**C**) apply to

*Figure 2 continued on next page*

*Figure 2 continued*

this panel as well. (G) Addition to the artificial perilymph in the top chamber of 1 mM EGTA, a $Ca^{2+}$ chelator that dissociates tip links, abolished the electrical response. The calibration bars in (C) apply to this panel as well. (H) When a superparamagnetic bead is tethered by DNA to a glass surface through DNA, thermal motion causes lateral deflections that are partially suppressed by the magnetic force $F$.

responses to stimuli of opposite polarities should have been identical. The artifacts, however, should have reversed signs in accordance with Faraday's law. To separate the transduction responses from artifacts, we therefore applied to the magnet successive voltage pulses of a constant magnitude but opposite polarities. By averaging the responses to these two stimuli we removed the artifacts and preserved only the responses owing to mechanoelectrical transduction (*Figure 2B*). The peak response in nine experiments was 38.7 ± 9.8 μV (mean ± standard deviation), roughly one-tenth the maximal value expected if every tip link were stimulated. That the responses to currents of opposite polarity were largely the same also indicated that the signals originated primarily from mechanoelectrical transduction rather than electromagnetic interference. As expected, gentamicin treatment revealed that the residual artifacts were equal and opposite and summed to zero.

Although the signals elicited by conventional hair-bundle deflection adapt rapidly to a plateau (*Eatock et al., 1987*; *Shepherd and Corey, 1994*), the responses elicited by protracted magnetic stimulation declined gradually over several seconds (*Figure 2C*). Conventional stimulation of a hair bundle is thought to tense each tip link and open the associated channels. Aided by the effect of $Ca^{2+}$ on the myosin molecules that secure the upper ends of the tip links, this tension then declines, resulting in adaptation of the electrical response (*Howard and Hudspeth, 1987*; *Assad et al., 1989*; *Hacohen et al., 1989*). Magnetic stimulation, in contrast, should pull the top ends of the tip links upward. This force not only opened transduction channels but evidently restrained the insertional plaques from descending as well. Recovery from the vertical offset of the plaques might explain why the electrical response relaxed far more slowly than during conventional adaptation. Because the magnitude of the response was sensitive to the strength of magnetic stimulation, the long plateau observed during protracted magnetic stimulation did not stem from saturation of the response.

Imperfect positioning of the saccular epithelium in the experimental chamber might have created a tilt between the preparation and the direction of the magnetic-field gradient. If this were the case, any component of magnetic force directed toward the tall edges of hair bundles would have excited a response by the conventional mechanism of bundle deflection. Although that form of stimulation would be difficult to reconcile with the protracted adaptation observed during extended magnetic stimulation, we nevertheless performed an additional experiment to eliminate this possibility and ensure that the measured potentials resulted from upward tension in the tip links. The preparation was positioned in a sample holder with a 10° tilt between the saccular epithelium and the plane of the magnet's pole piece (*Figure 2D*). After tethering superparamagnetic beads as usual, we scraped away most of the beads with an eyelash, leaving beads and intact hair bundles only in the segment of the epithelium opposite the saccular nerve (*Figure 2E*). Throughout that region the hair bundles display a common direction of excitability away from the nerve. In the tilted configuration any magnetic force tangent to the epithelial surface would have pulled the hair bundles toward their short edges and would therefore have countered the conventional electrical response. Even under these conditions we observed a transduction current of the usual polarity but of diminished magnitude owing to the removal of most tethered beads (*Figure 2F*). As expected, the response vanished when the remaining beads had been scraped away.

Attachment of the magnetic tethers to cadherins on hair-cell surfaces rather than to fully formed tip links might elicit extraneous responses independent of the transduction process. To eliminate this possibility, we dissociated tip links by treatment with 1 mM EGTA. Under these conditions the summed response fell to zero (*Figure 2G*), indicating that the responses stemmed from traction on tip links.

In order to quantify the stimulus, we calibrated the force exerted by the magnet on a superparamagnetic bead attached to a glass surface through a DNA tether (*Gosse and Croquette, 2002*). The lateral Brownian fluctuations of the tethered bead were measured at 250 frames per second with a video camera (Redlake MotionScope 2000S, DEL Imaging Systems). When the magnetic field was applied with the pole piece 4.1 mm distant, the axial force on the bead partially suppressed its

lateral Brownian fluctuations. The equipartition theorem relates the axial pulling force $F$ and the magnitude of lateral fluctuations of the bead's position $\langle \delta X^2 \rangle$:

$$F = kT \left( \frac{L}{\langle \delta X^2 \rangle} \right), \tag{1}$$

in which $k$ is the Boltzmann constant, $T$ the temperature, and $L$ the time-averaged end-to-end length of the DNA molecule (*Figure 2H*). The wormlike-chain model for the extension of DNA under a load then relates $F$ to $L$ (*Bustamante et al., 1994*):

$$F = \left( \frac{kT}{P} \right) \left[ \frac{1}{4(1 - L/L_0)^2} + \frac{L}{L_0} - \frac{1}{4} \right], \tag{2}$$

in which $P$ is the DNA molecule's persistence length of 50 nm and $L_0$ is its contour length of 3000 base pairs or 1.02 μm. By measuring the lateral displacement $\langle \delta X^2 \rangle$ and numerically solving *Equations (1) and (2)* for 16 determinations, we estimated a magnetic force at the position of the specimen of 0.48 ± 0.07 pN (mean ± standard deviation).

## Discussion

Because the inaccessibility of tip links obstructs their direct manipulation, mechanical stimulation of hair cells has heretofore been restricted to deflections of entire hair bundles. By overcoming this limitation, our experiments have demonstrated that upward force on tip links generates transduction currents independent of hair-bundle deflection. As expected, these responses vanish in the presence of a channel blocker. The observed responses accord well with the values expected on the basis of the calibrated magnetic force on tethered beads and the gating-spring theory for transduction by hair cells.

More than three decades after the postulation that tension in gating springs within the hair bundle opens transduction channels (*Corey and Hudspeth, 1983a*) and the subsequent suggestion that tip links constitute those elements (*Pickles et al., 1984*), the present observations provide direct support for both hypotheses. The technical approach introduced here additionally affords a means of investigating mechanoelectrical transduction at the scale of individual tip links and channels.

## Materials and methods

### Polyclonal antiserum against the bullfrog's CDH23

Wishing to develop an antiserum against an epitope of the bullfrog's CDH23 near the upper insertion of the tip link, we selected the region corresponding to the $Ca^{2+}$-binding linker between extracellular cadherin domains EC25 and EC26 of the murine protein (UniProtKB/Swiss-Prot: Q99PF4.2). This region was selected on the basis of its probable accessibility and antigenicity and because it occurs near the uppermost of CDH23's 27 extracellular cadherin domains. Using primers based on the DNA sequence for the zebrafish's *cdh23* (NM_214809), we obtained a product by RT-PCR of the RNA isolated from a single bullfrog's sacculus (Superscript III One-Step RT-PCR System, Invitrogen). This cDNA was inserted into the vector pCR2.1-TOPO and sequenced to confirm its identity. The corresponding polypeptide DDNEPIFVRPPRGA was synthesized with the addition of a cysteine residue at its amino terminal and with amidation of its carboxyl terminal. Following the immunization of two rabbits (Covance), the resultant serum (RU1793) was affinity-purified against the immunizing peptide.

### Anti-fluorescein-coated superparamagnetic beads

After 250 μL of 1 μm-diameter carboxylate-functionalized superparamagnetic beads (MyOne, Life Technologies) had been washed twice in 15 mM 2-(*N*-morpholino) ethanesulfonate at pH 6, they were resuspended in 45 μL of the same solution. Following the addition of 5 μL of 52 mM 1-ethyl-3-(3-dimethylaminopropyl) carbodiimide (Thermo Scientific) the beads were tumbled for 30 min. The beads were then sedimented with a magnet and the solution was decanted.

After 75 µL of rabbit anti-fluorescein (Thermo Scientific) had been diluted into 100 µL of the foregoing buffer solution, 100 µL of the product was added to the beads and tumbling was continued for 16 hr. Following two rinses in $Ca^{2+}$- and glucose-free artificial perilymph, the beads were resuspended in 125 µL of that solution.

## Tethering of superparamagnetic beads to tip links

Double-stranded, 3 kb DNA molecules modified at their opposite ends with biotin and fluorescein were produced by PCRs from the plasmid PGBKT7; the primers were 5′-biotin·dT-TTCTGGCAAC-CAAACCCATACATCG-3′ and 5′-fluorescein·dT-TTCTATGAAAGGTTGGGCTTCGGA-3′ (IDT DNA). Each such DNA molecule has a contour length of about 1 µm but displays a radius of gyration about one-third that great when unloaded. After the DNA had been eluted into 10 mM tris (hydroxymethyl) aminomethane at pH 8 and concentrated to 700 ng/µL, 4 µL of the resultant solution was mixed with an equal volume of 1 mg/mL streptavidin (Sigma) in $Ca^{2+}$- and glucose-free artificial perilymph and incubated at room temperature for 30 min. Following the addition of 20 µL of functionalized superparamagnetic beads to this mixture and tumbling for 30 min, the beads were sedimented with a magnet, washed five times with blocking solution consisting of 10% normal goat serum (Jackson Research) in artificial perilymph, and resuspended in 80 µL of blocking solution. Pipetting was conducted gently to prevent rupture of the DNA tethers by excessive shear.

After the apical surface of a saccular macula mounted in a two-compartment chamber had been exposed to blocking solution for 10 min, the solution in the top chamber was replaced for 10 min with anti-CDH23 diluted 1:100 in blocking solution. That chamber was rinsed four times with blocking solution and biotinylated goat anti-rabbit secondary antibody (Life Technology) was added for 10 min at a 1:100 dilution. The top chamber was then rinsed five times with blocking solution and 40 µL of DNA-tethered superparamagnetic beads was gently added to the chamber. Following incubation for 20 min, excess beads were gently washed away with blocking solution.

## Stimulation and recording

All procedures were approved by the University's Institutional Animal Care and Use Committee. Sacculi were dissected from adult bullfrogs (*Rana catesbeiana*) of both sexes into oxygenated artificial perilymph containing 114 mM $Na^+$, 2 mM $K^+$, 0.25 mM $Ca^{2+}$, 116 mM $Cl^-$, 5 mM HEPES, and 3 mM D-glucose. After the removal of otoconia, each saccular macula was sealed with *n*-butyl cyanoacrylate (Vetbond, 3M) over a 1 mm-diameter hole in a 12 mm-diameter plastic disk to form the partition in a two-compartment recording chamber. The apical surface was exposed to 67 µg/mL of protease (type XXIV; Sigma) for 30 min at 22°C to loosen the otolithic membrane, which was removed with an eyelash. Both compartments of the recording chamber were then filled with artificial perilymph.

The recording chamber was centered 4.1 mm below the pole piece of a solenoidal electromagnet (E-40-300-15, Magnetic Sensor Systems) that was actuated by a power amplifier (PA-119, Labworks). Experimental control, stimulation, and recording were effected with a data-acquisition card (PCIe-6353, National Instruments) and data were analyzed with MATLAB. The potential difference across the tissue was measured with a directly coupled extracellular amplifier (EXT-02F, NPI Electronic) with a gain of 200X and a low-pass cutoff frequency of 1.3 kHz. The response was recorded differentially and is presented as the potential in the upper chamber referenced to that in the lower. Each record represents the average of eight repetitions of each stimulus polarity and was smoothed to a maximal frequency of 1 kHz by rolling-window averaging.

## Tethering of superparamagnetic beads for force calibration

A cover glass was washed with HCl, coated for 30 min with 1 mg/mL concanavalin A, and rinsed with water. The treated cover glass was sequentially exposed to anti-CDH23 diluted 1:100 in PBS, washed with blocking solution, treated with biotinylated secondary antibody diluted 1:100 in blocking solution, washed with blocking solution, treated with DNA functionalized at either end with biotin and fluorescein at a concentration of 20 pM, washed with blocking solution, treated with superparamagnetic beads functionalized with anti-fluorescein diluted 1:10 in blocking solution, and finally washed with blocking solution.

## Ab initio estimation of magnetic stimuli

We compared our calibration of magnetic force with ab initio expectations, based on the nominal magnetic properties of the bead and our measurements of the strength and gradient of the magnetic field.

In the linear range of magnetization of a bead of volume $V$ owing to a flux density $\mathbf{B}$, for which the magnetization $\mathbf{M}$ is related to the field strength $\mathbf{H}$ by the volume susceptibility $\chi$, the induced magnetic dipole moment $\mathbf{m}$ is given by

$$\mathbf{m} = V\mathbf{M} = V\chi\mathbf{H} = \left(\frac{V\chi}{\mu_0}\right)\mathbf{B}. \tag{3}$$

Because the potential energy $U$ of a magnetic dipole is $U = -\mathbf{m}\cdot\mathbf{B}/2$, the vertical component of force $\mathbf{F}_Z$ on the dipole is

$$\mathbf{F}_Z = -\vec{\nabla}_Z U = \frac{1}{2}\vec{\nabla}_Z\left(\frac{V\chi}{\mu_0}|\mathbf{B}|^2\right) = \left(\frac{V\chi}{\mu_0}\right)|\mathbf{B}|\frac{\partial|\mathbf{B}|}{\partial Z}. \tag{4}$$

Here the positive value reflects the fact that, regardless of the direction of current through the electromagnet, a superparamagnetic bead is pulled into an increasing magnetic field, in this instance upward.

Using a Hall-effect magnetometer (OHS3150U; TT Electronics/Optek Technology), we measured a magnetic field strength of 49 mT (490 G) at a distance of 4.1 mm from the magnet's pole piece and along its center axis. Measurements made after slight deflections of the detector yielded an estimated gradient of 1.6 T·m$^{-1}$. Substituting these values and the manufacturer's stated volume susceptibility of 1.4 into the foregoing relation leads to an estimate for the vertical force on each bead of 0.05 pN, lower than our calibrated value of 0.48 pN. Ab initio calculations provide only a lower limit to the magnetic force, however, and require scaling by an empirically determined factor to accord with measured values (*Lipfert et al., 2009*; *Shevkoplyas et al., 2007*).

## Estimation of expected electrical responses

The gating-spring theory of transduction (*Corey and Hudspeth, 1983a*; *Hudspeth, 1992*; *Markin and Hudspeth, 1995*) asserts that the change in energy $\Delta E$ associated with the opening of an individual transduction channel is

$$\Delta E_{C\rightarrow O} = \Delta E^\phi - \kappa d(\gamma X + x_C - d/2), \tag{5}$$

in which $\Delta E^\phi$ is the intrinsic component of gating energy in the absence of tip-link tension, $k$ the gating-spring stiffness, $d$ the swing of a transduction channel's gate, $\gamma$ the bundle's geometrical gain, and $x_C$ the elongation of the gating spring when attached to a closed channel in a resting bundle. For an undisturbed hair bundle, the displacement $X$ is zero, so each channel's open probability $P_0$ is

$$P_O = \frac{1}{1 + e^{\Delta E_{C\rightarrow O}/kT}} = \frac{1}{1 + e^{[E^\phi - \kappa d(x_C - d/2)]/kT}}, \tag{6}$$

in which $k$ is the Boltzmann constant and $T$ the temperature. If the gating-spring tension is increased, for example by a magnetically induced force $F$, the expected change in open probability is

$$\frac{dP_O}{dx_C} = P_O(1 - P_O)\left(\frac{\kappa d}{kT}\right). \tag{7}$$

In the saline solution used for our experiments, $P_0 \approx 0.2$, $k \approx 500\,\mu\text{N}\cdot\text{m}^{-1}$, and $d \approx 5\,\text{nm}$ (*Corey and Hudspeth, 1983b*; *Howard and Hudspeth, 1988*; *Martin et al., 2000*), so we anticipate a sensitivity of $dP_0/dx_C \approx 10^8\,\text{m}^{-1}$. For a tip link of the stated stiffness, this value represents a responsiveness to force of $dP_0/dF \approx 2\cdot10^{11}\,\text{N}^{-1}$. According to our force calibrations, magnetic stimulation induces a force of $\Delta F = 0.48\,\text{pN}$, which would thus be expected to enhance the open probability by $\Delta P_0 \approx 0.10$. The *ab initio* calculation estimates $\Delta F = 0.05\,\text{pN}$, which would imply $\Delta P_0 \approx 0.01$.

There are two ways to estimate the maximal transepithelial current that might be evoked by activating all the transduction channels. First, a cluster of hair bundles representing about one-seventh of the saccular complement produces a peak transepithelial response near 500 μV during conventional

stimulation (*Corey and Hudspeth, 1983b*), so activating every cell in the sensory epithelium would yield about 3500 µV. The second approach is to consider that an individual hair cell from the frog's sacculus typically produces a peak transduction current of 200 pA (*Gillespie and Hudspeth, 1993*; *Cheung and Corey, 2006*), so the 2000 or so cells in a medium-sized sacculus (*Jacobs and Hudspeth, 1990*) would yield a total current of about 400 nA. Returning across an epithelium of resistance 10 kΩ, that current would yield a transepithelial signal of 4000 µV, a value in good agreement with the first calculation. If magnetic stimulation were to increase the open probability of all the preparation's transduction channels by 10% as calculated from our calibration of the magnetic force, the magnetic stimuli would be expected to produce signals of 350–400 µV at most. The value would be approximately one-tenth as great on the basis of *ab initio* calculations. Because only a small fraction of the tip links are attached to beads, however, the actual responses should be still smaller.

## Acknowledgements

The authors thank T Bartsch for assistance in the magnetic calibration experiments and C Kirst and members of our research group for comments on the manuscript. A Basu was a Simons Foundation Fellow of the Life Sciences Research Foundation. M Vologodskaia and B Fabella were supported by Howard Hughes Medical Institute, of which AJ Hudspeth is an Investigator.

## Additional information

### Funding

| Funder | Author |
| --- | --- |
| Howard Hughes Medical Institute | Aakash Basu<br>Samuel Lagier<br>Maria Vologodskaia<br>Brian A Fabella<br>AJ Hudspeth |

The funders had no role in study design, data collection and interpretation, or the decision to submit the work for publication.

### Author contributions

AB, Conception and design, Acquisition of data, Analysis and interpretation of data, Drafting or revising the article; SL, Drafting or revising the article, Contributed unpublished essential data or reagents; MV, Acquisition of data, Drafting or revising the article, Contributed unpublished essential data or reagents; BAF, Conception and design, Acquisition of data, Drafting or revising the article; AJH, Conception and design, Analysis and interpretation of data, Drafting or revising the article

### Author ORCIDs

AJ Hudspeth, http://orcid.org/0000-0002-0295-1323

### Ethics

Animal experimentation: All procedures were approved by the University's Institutional Animal Care and Use Committee under protocol 13665. Animals were sacrificed by dual pithing after anesthesia by immersion in ethyl-3-aminobenzoate methanesulfonic acid or 4-allyl-2-methoxyphenol.

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
