## [Decision Letter]

Thank you for submitting your article "Direct mechanical stimulation of tip links in hair cells through DNA tethers" for consideration by *eLife*. Your article has been favorably evaluated by Richard Aldrich (Senior editor) and three reviewers, one of whom is a member of our Board of Reviewing Editors.

The reviewers have discussed the reviews with one another and the Reviewing Editor has drafted this decision to help you prepare a revised submission.

Summary:

The manuscript reports that transepithelial currents can be evoked by directly pulling on CDH23 via an antibody using a magnetic field. This is an important and new result if the authors can convincingly show that in their experiments they are pulling on tip links (rather than CDH23 molecules not assembled into tip links) and that the force is not moving the hair bundle. While the work presented in the manuscript is strong, it is not yet fully convincing and a specific additional experiment is needed to show that the stimulation due to upward forces indeed depends on the presence of intact tip links on which the force is applied.

Essential revisions:

The authors should demonstrate that the upward force is exerted on tip links and not on Cadherin 23 molecules that are present on the cell surface but are not part of a tip-link spanning neighbouring stereocilia. This can be shown using BAPTA treatment which severs the tip-links. As a result of this treatment, the direct hair bundle stimulation should no longer be observed as there are not tip links left to which the force can be applied.

A second point concerns the calibration and the estimation of the applied forces. This is presented in the supplement and is not very clearly presented. As this is an important part of the work it should be clearly discussed in the main text.

---

## [Author Response]

Essential revisions:

The authors should demonstrate that the upward force is exerted on tip links and not on Cadherin 23 molecules that are present on the cell surface but are not part of a tip-link spanning neighbouring stereocilia. This can be shown using BAPTA treatment which severs the tip-links. As a result of this treatment, the direct hair bundle stimulation should no longer be observed as there are not tip links left to which the force can be applied.

We agree with the reviewers that this is an important control. We have added an illustration (Figure 2) portraying an equivalent experiment that we had already performed: the response to magnetic stimulation drops to zero when the preparation is treated with 1 mM EGTA. Like other Ca^2+^ chelators, this substance dissociates tip links and arrests mechanoelectrical transduction. We have also included the following in the main text: "Attachment of the magnetic tethers to cadherins on hair-cell surfaces rather than to fully formed tip links might elicit extraneous responses independent of the transduction process. To eliminate this possibility, we dissociated tip links by treatment with 1 mM EGTA. Under these conditions the summed response fell to zero (Figure 2), indicating that the responses stemmed from traction on tip links."

A second point concerns the calibration and the estimation of the applied forces. This is presented in the supplement and is not very clearly presented. As this is an important part of the work it should be clearly discussed in the main text.

In response to this comment, we have transferred the section describing the force calibration to the main text. We have also added detail and attempted to be more explicit about the procedure. The following has now been added to the main text:

"In order to quantify the stimulus, we calibrated the force exerted by the magnet on a superparamagnetic particle attached to a glass surface through a DNA tether (Gosse and Croquette, 2002). […] By measuring the lateral displacement *<δx^2^>* and numerically solving [Disp-formula equ1 equ2] for 16 determinations, we estimated a magnetic force at the position of the specimen of 0.48 ± 0.07 pN (mean ± standard deviation)."

We have retained the ab initio estimation for force from the nominal properties of the bead and the field strength and gradient in the Materials and methods. We have also conducted additional calibration experiments and report the results:

"Using a Hall-effect magnetometer (OHS3150U; TT Electronics/Optek Technology), we measured a magnetic field strength of 49 mT (490 G) at a distance of 4.1 mm from the magnet's pole piece and along its center axis. […] An ab initio calculation provides only a lower estimate of the magnetic force, however, and requires scaling by an empirically determined factor to accord with measured values (Lipfert et al., 2009; Shevkoplyas et al., 2007).”